# Transposon silencing in the *Drosophila* female germline is essential for genome stability in progeny embryos

Zeljko Durdevic[1], Ramesh S Pillai[2], Anne Ephrussi[1]

The Piwi-interacting RNA pathway functions in transposon control in the germline of metazoans. The conserved RNA helicase Vasa is an essential Piwi-interacting RNA pathway component, but has additional important developmental functions. Here, we address the importance of Vasa-dependent transposon control in the *Drosophila* female germline and early embryos. We find that transient loss of *vasa* expression during early oogenesis leads to transposon up-regulation in supporting nurse cells of the fly egg-chamber. We show that elevated transposon levels have dramatic consequences, as de-repressed transposons accumulate in the oocyte where they cause DNA damage. We find that suppression of Chk2-mediated DNA damage signaling in *vasa* mutant females restores oogenesis and egg production. Damaged DNA and up-regulated transposons are transmitted from the mother to the embryos, which sustain severe nuclear defects and arrest development. Our findings reveal that the Vasa-dependent protection against selfish genetic elements in the nuage of nurse cell is essential to prevent DNA damage–induced arrest of embryonic development.

## Introduction

Transposons and other selfish genetic elements are found in all eukaryotes and comprise a large fraction of their genomes. Although transposons are thought to be beneficial in driving evolution (Levin & Moran, 2011), their mobilization in the germline can compromise genome integrity with deleterious consequences: insertional mutagenesis reduces the fitness of the progeny, and loss of germ cell integrity causes sterility. Therefore, it is of great importance for sexually reproducing organisms to firmly control transposon activity in their germ cells. Metazoans have evolved a germline-specific mechanism that, by relying on the activity of Piwi family proteins and their associated Piwi-interacting RNAs (piRNAs), suppresses mobile elements.

*Drosophila* harbors three PIWI proteins: Piwi, Aubergine (Aub), and Argonaute 3 (Ago3), which, guided by piRNAs, silence transposons

at the transcriptional and posttranscriptional levels (reviewed in Guzzardo et al [2013]). Besides PIWI proteins, other factors such as Tudor domain proteins and RNA helicases are involved in piRNA biogenesis and transposon silencing. Mutations in most piRNA pathway genes in *Drosophila* females cause transposon up-regulation that leads to an arrest of oogenesis. This effect can be rescued by suppression of the DNA damage checkpoint proteins of the ATR/Chk2 pathway (Chen et al, 2007; Klattenhoff et al, 2007; Pane et al, 2007). By contrast, inhibition of DNA damage signaling cannot restore embryonic development (Chen et al, 2007; Klattenhoff et al, 2007; Pane et al, 2007). Recent studies suggest that PIWI proteins might have additional roles during early embryogenesis independent of DNA damage signaling (Khurana et al, 2010; Mani et al, 2014). However, functions of the piRNA pathway during early embryonic development remain poorly understood.

One of the essential piRNA pathway factors with an important role in development is the highly conserved RNA helicase Vasa. First identified in *Drosophila* as a maternal-effect gene (Schüpbach & Wieschaus, 1986; Hay et al, 1988; Lasko & Ashburner, 1990), *vasa* (*vas*) was subsequently shown to function in various cellular and developmental processes (reviewed in Lasko [2013]). In the *Drosophila* female germline, Vasa accumulates in two different cytoplasmic electron-dense structures: the pole (or germ) plasm at the posterior pole of the oocyte, and the nuage, the perinuclear region of nurse cells. In the pole plasm, Vasa interacts with the pole plasm–inducer Oskar (Osk) (Markussen et al, 1995; Jeske et al, 2015) and ensures accumulation of different proteins and mRNAs that determine primordial germ cell (PGC) formation and embryonic patterning (Hay et al, 1988; Lasko & Ashburner, 1990). In the nuage, Vasa is required for the assembly of the nuage itself (Liang et al, 1994; Malone et al, 2009) and facilitates the transfer of transposon RNA intermediates from Aub to Ago3, driving the piRNA amplification cycle and piRNA-mediated transposon silencing (Xiol et al, 2014; Nishida et al, 2015). As Vasa's involvement in many cellular processes renders it difficult to analyze its functions in each process individually, it remains unknown whether Vasa's functions in development and in the piRNA pathway are linked or independent.

In this study, we address the role of Vasa in transposon control in *Drosophila* development. We find that failure to suppress transposons in the nuage of nurse cells causes DNA double-strand

---

[1]Developmental Biology Unit, European Molecular Biology Laboratory, Heidelberg, Germany    [2]Department of Molecular Biology, University of Geneva, Geneva, Switzerland

Correspondence: ephrussi@embl.de

breaks (DSBs), severe nuclear defects, and lethality of progeny embryos. Even transient interruption of Vasa expression in early oogenesis de-represses transposons and impairs embryo viability. Depletion of the *Drosophila* Chk2 ortholog *maternal nuclear kinase* (*mnk*) restores oogenesis in *vas* mutants, but does not suppress defects in transposon silencing or DSB-induced nuclear damage and embryonic lethality. We show that up-regulated transposons invade the maternal genome, inducing DNA DSBs that, together with transposon RNAs and proteins, are maternally transmitted and consequently cause embryogenesis arrest. Our study thus demonstrates that Vasa function in the nuage of *Drosophila* nurse cells is essential to maintain genome integrity in both the oocyte and progeny embryos, ensuring normal embryonic development.

# Results

## Vasa-dependent transposon control is not essential for oogenesis

Vasa is required for piRNA biogenesis and transposon silencing in *Drosophila*, as in *vas* mutants piRNAs are absent and transposons are up-regulated (Vagin et al, 2004; Malone et al, 2009; Zhang et al, 2012; Czech et al, 2013; Handler et al, 2013). To investigate the importance of transposon control in *Drosophila* development, we expressed WT GFP-Vasa fusion protein (GFP-Vas$^{WT}$; Fig S1A) in the female germline of loss-of-function (*vas*$^{D1/D1}$) *vas* flies using two promoters with distinct expression patterns (Fig S1B and C): the *vas* promoter is active at all stages of oogenesis, whereas the *nos* promoter is attenuated between stages 2 and 6 (Fig S1B and C).

We first assessed the ability of GFP-Vas$^{WT}$ fusion protein to promote transposon silencing in the female germline, and examined the effect of GFP-Vas$^{WT}$ on the level of expression of several transposons in *vas* mutant ovaries. We chose the LTR retrotransposons *burdock* and *blood*, and the non-LTR retrotransposon *HeT-A*, which were previously reported to be up-regulated upon Vasa depletion (Vagin et al, 2004; Czech et al, 2013). The LTR retrotransposon *gypsy*, which belongs to the so-called somatic group of transposons and is not affected by Vasa depletion, served as a negative control (Czech et al, 2013). Loss-of-function *vas*$^{D1/D1}$ ovaries contained elevated levels of *burdock*, *blood*, and *HeT-A* RNA (Fig 1A). Remarkably, silencing of transposons by GFP-Vas$^{WT}$ in *vas*$^{D1/D1}$ flies depended on which Gal4 driver was used (Fig S1B and C): When driven by *nos-Gal4*, GFP-Vas$^{WT}$ had no effect on transposon levels, whereas when driven by *vas-Gal4*, it led to the re-silencing of transposons (Fig 1A). This differential effect presumably reflects the stages of oogenesis at which the *nos* and *vas* promoters are active, and suggests that lack of Vasa between stages 2 and 6 of oogenesis (Fig S1B) leads to transposon de-repression. Importantly, independent of Gal4 driver used, expression of GFP-Vas$^{WT}$ restored oogenesis (Figs 1B and S1D) and egg-laying (Fig S1E and F). The fact that in spite of transposon up-regulation oogenesis and egg-laying rates were largely restored in *vas*$^{D1/D1}$ flies (Fig 1A, indicated by + and – and Fig S1D–F) is consistent with the notion that transposon activation affects but does not completely block oogenesis unless the level of activation is so high as to cause its arrest.

## Loss of Vasa during early oogenesis affects viability of progeny embryos

Concentration of Vasa protein at the posterior pole of the embryo is essential for PGC and abdomen formation during embryogenesis (Schüpbach & Wieschaus, 1986; Hay et al, 1988; Lasko & Ashburner, 1990). We analyzed the number of PGC-positive embryos and the hatching rate of eggs produced by *vas*$^{D1/D1}$ flies expressing GFP-Vas$^{WT}$ either under control of the *nos* or the *vas* promoter (*vas*$^{D1/D1}$; *nos-Gal4>GFP-vas*$^{WT}$ and *vas*$^{D1/D1}$; *vas-Gal4>GFP-vas*$^{WT}$ embryos). Embryos from *vas*$^{D1/D1}$ mutant flies could not be included in these and all the other experiments on embryos, as *vas*$^{D1/D1}$ females arrest oogenesis early and do not lay eggs. PGC formation was restored in approximately 50% of *vas*$^{D1/D1}$; *nos-Gal4>GFP-vas*$^{WT}$ and *vas*$^{D1/D1}$; *vas-Gal4>GFP-vas*$^{WT}$ embryos (Fig 1C) (Table S1). However, DAPI staining revealed nuclear damage in some *vas*$^{D1/D1}$; *nos-Gal4>GFP-vas*$^{WT}$ embryos (see below), which we excluded from the quantification.

Expression of GFP-Vas$^{WT}$ also partially rescued the hatching of eggs produced by *vas*$^{D1/D1}$ flies (Fig 1D). However, expression of GFP-Vas$^{WT}$ led to a significantly lower hatching rate in *vas*$^{D1/D1}$; *nos-Gal4>GFP-vas*$^{WT}$ than in *vas*$^{D1/D1}$; *vas-Gal4>GFP-vas*$^{WT}$ flies (Fig 1D) (Table S2). Expression of GFP-Vas$^{WT}$ in heterozygous loss-of-function *vas*$^{D1/Q7}$ females led to a low hatching rate similar to *vas*$^{D1/D1}$ (Fig S2A and B) (Table S3), excluding a possible secondary mutation as the cause of the low hatching rate. The fact that in spite of comparable GFP-Vas$^{WT}$ levels (Fig S2C), the hatching rate of *vas*$^{D1/D1}$; *vas-Gal4>GFP-vas*$^{WT}$ embryos was higher than that of *vas*$^{D1/D1}$; *nos-Gal4>GFP-vas*$^{WT}$ embryos, suggests that transient loss of *vas* expression during early oogenesis impairs viability of progeny embryos (Fig 1D).

## Elevated transposon levels cause DNA and nuclear damage in progeny embryos

Elevated transposon activity leads to DNA damage and ultimately to cell death. During our analysis of PGC formation, we observed nuclear damage in a considerable fraction of *vas*$^{D1/D1}$; *nos-Gal4>GFP-vas*$^{WT}$ embryos. Quantification of embryos containing nuclei of aberrant nuclear morphology (Fig 2A, lower panel) compared with the nuclei of WT embryos (Fig 2A, upper panel) revealed a high proportion of such nuclear defects among *vas*$^{D1/D1}$; *nos-Gal4>GFP-vas*$^{WT}$ embryos (Fig 2A). Transposon mobilization causes DSBs in genomic DNA that are marked by the incorporation of a phosphorylated form of the H2A variant ($\gamma$H2Av), a histone H2A variant involved in DNA DSB repair. Analysis of $\gamma$H2Av occurrence showed that embryos displaying nuclear damage were $\gamma$H2Av-positive (Fig 2B), indicating that DNA DSBs cause nuclear defects. The levels of $\gamma$H2Av were higher in *vas*$^{D1/D1}$; *nos-Gal4>GFP-vas*$^{WT}$ embryos compared with WT and *vas*$^{D1/D1}$; *vas-Gal4>GFP-vas*$^{WT}$ (Fig 2C) (Table S4).

The correlation between high levels of transposon expression during oogenesis (Fig 1A, *nos-Gal4*-driven) and a high frequency of nuclear damage and DSBs in *vas*$^{D1/D1}$; *nos-Gal4>GFP-vas*$^{WT}$ embryos (Fig 2A–C) suggested that maternally transmitted transposons cause embryonic lethality. To test this, we compared transposon RNA levels in embryos of *vas*$^{D1/D1}$; *nos-Gal4>GFP-vas*$^{WT}$ and *vas*$^{D1/D1}$; *vas-Gal4>GFP-vas*$^{WT}$ flies, in which transposon RNAs are up- and down-regulated, respectively (Fig 1A). Levels of maternally transmitted transposon RNA were significantly higher in *vas*$^{D1/D1}$; *nos-Gal4>GFP-vas*$^{WT}$

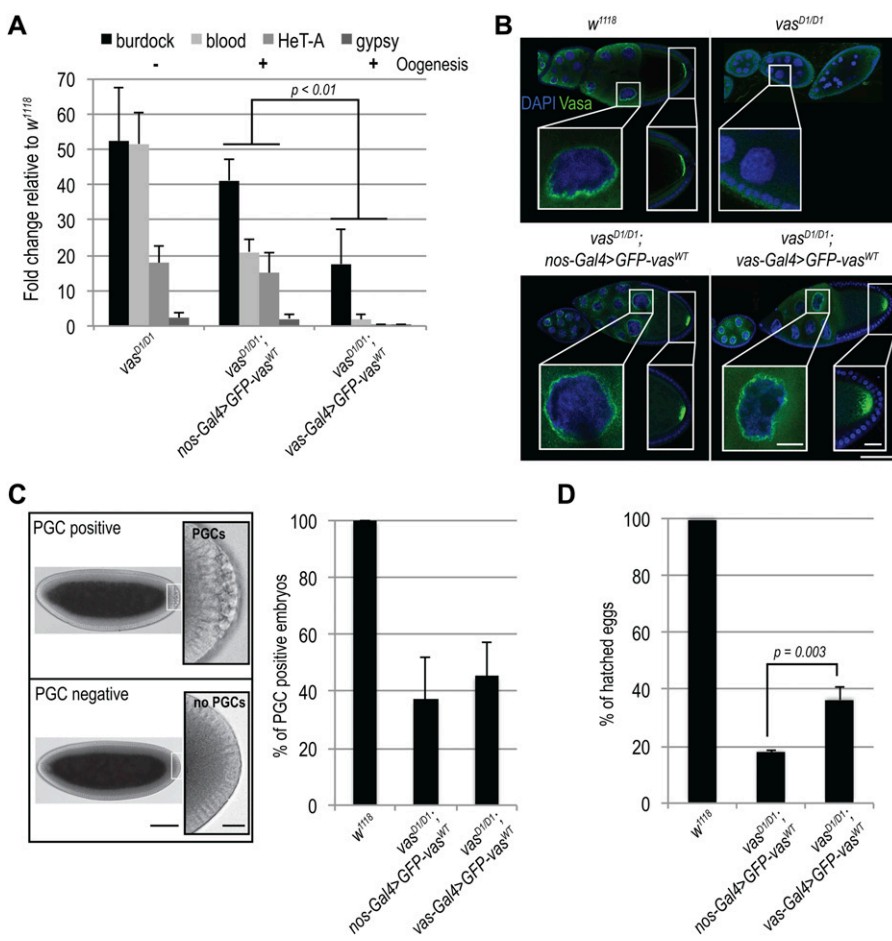

**Figure 1. Silencing of transposon RNAs during oogenesis is essential for embryonic development.**
**(A)** qPCR analysis of LTR transposons *burdock*, *blood*, and *gypsy* and non-LTR transposon HeT-A RNAs in *vas$^{D1/D1}$*, *vas$^{D1/D1}$*; *nos-Gal4>GFP-vas$^{WT}$ and vas$^{D1/D1}$*; *vas-Gal4>GFP-vas$^{WT}$*, ovaries. Expression level of transposons in WT (*w$^{1118}$*) was set to one and normalized to rp49 mRNA in individual experiments. Error bars represent the standard deviation among three biological replicates. *P*-values were determined by *t* test. *P*-values for *burdock* (0.006), *blood* (0.0002), and HeT-A (0.0007) were lower than 0.01 (indicated in the chart), whereas *gypsy* levels were not significantly different (*P* = 0.5). Oogenesis completion is indicated with + and –. **(B)** Immunohistochemical detection of Vasa in WT (*w$^{1118}$*) and *vas$^{D1/D1}$* flies (upper panel), and GFP signal of GFP-Vas$^{WT}$ fusion protein in *vas$^{D1/D1}$*; *nos-Gal4>GFP-vas$^{WT}$ and va$^{D1D/D1}$*; *vas-Gal4>GFP-vas$^{WT}$* flies (lower panel). Insets show enlarged images of nuage and oocyte posterior pole. Scale bars indicate 50 µm (egg-chambers) and 10 µm (nuage and pole plasm). **(C)** Quantification of PGC-positive embryos produced by WT (*w$^{1118}$*), *vas$^{D1/D1}$*; *nos-Gal4>GFP-vas$^{WT}$*, and *vas$^{D1/D1}$*; *vas-Gal4>GFP-vas$^{WT}$* flies. Error bars represent the standard deviation among three biological replicates (Table S1). Panel (left) shows PGC-positive embryo (top) and PGC-negative embryo (bottom). Scale bars indicate 100 µm (embryo) and 5 µm (PGCs). **(D)** Hatching rates of eggs laid by WT (*w$^{1118}$*), *vas$^{D1/D1}$*; *nos-Gal4>GFP-vas$^{WT}$*, and *vas$^{D1/D1}$*; *vas-Gal4>GFP-vas$^{WT}$* flies. Error bars represent the standard deviation among three biological replicates (Table S2). *P*-value was determined by *t* test.

embryos (Fig 2D) suggesting that the increased lethality observed in *vas$^{D1/D1}$*; *nos-Gal4>GFP-vas$^{WT}$* embryos is due to DNA damage (Fig 2A–C) caused by the high levels of maternally transmitted transposon RNAs (Fig 2D).

One of the up-regulated transposons in *vas* mutants is HeT-A, whose RNA and protein expression is strongly de-repressed in piRNA pathway mutant ovaries (Aravin et al, 2001; Vagin et al, 2006; Zhang et al, 2014; Lopez-Panades et al, 2015). Analysis of HeT-A/Gag protein expression in 0–1 h old embryos showed that the levels of HeT-A/Gag were much higher in *vas$^{D1/D1}$*; *nos-Gal4>GFP-vas$^{WT}$* than in *vas$^{D1/D1}$*; *vas-Gal4>GFP-vas$^{WT}$* embryos (Fig 2E). In addition, we stained embryos with antibodies against HeT-A/Gag protein and observed that in cellularized WT embryos, HeT-A localized in distinct perinuclear foci (Figs 3A, panel a and S3A, panel a), as previously described for HeT-A/Gag-HA-FLAG fusion protein (Olovnikov et al, 2016). In *vas$^{D1/D1}$*; *nos-Gal4>GFP-vas$^{WT}$* embryos displaying nuclear damage, HeT-A protein accumulated in large foci throughout the embryo (Figs 3A, panel b and S3A, panel b), whereas embryos of the same genotype lacking nuclear damage showed a WT distribution of the protein (Figs 3A, panel c and S3A, panel c). Finally, HeT-A/Gag displayed WT localization in *vas$^{D1/D1}$*; *vas-Gal4>GFP-vas$^{WT}$* embryos (Figs 3A, panel d and S3A, panel d). Altogether, these results show that up-regulation of transposon mRNAs and proteins during oogenesis results in their maternal transmission to the progeny, where they cause DSBs, nuclear damage, and arrest of embryogenesis.

## Chk2 mutation restores oogenesis but not transposon silencing and embryogenesis in *vas* mutants

To test genetically whether DNA damage signaling contributes to the oogenesis arrest of *vas* loss-of-function mutants (Schüpbach & Wieschaus, 1986; Hay et al, 1988; Lasko & Ashburner, 1990) (Fig 1A), we introduced the *mnk$^{P6}$* loss-of-function allele into the *vas$^{D1}$* background. Genetic removal of *mnk* (Fig S3B) suppressed the oogenesis arrest of *vas$^{D1/D1}$* mutants and partially rescued their egg laying (Figs S3C and S4A). Importantly, *mnk$^{P6/P6}$* single mutants expressed Vasa at WT levels and, as expected, the protein was not detected in *vas$^{D1/D1}$*, *mnk$^{P6/P6}$* double mutants (Fig S2C). Taken together, these findings demonstrate that the oogenesis arrest of loss-of-function *vas* mutants results from activation of the Chk2-mediated DNA damage-signaling checkpoint.

Although removal of *mnk* allowed oogenesis progression, it did not reduce transposon levels in *vas$^{D1/D1}$*, *mnk$^{P6/P6}$* ovaries, and the eggs laid failed to hatch (Figs 3B and S4B) (Table S5). Further analysis revealed that *vas$^{D1/D1}$*, *mnk$^{P6/P6}$* early embryos contained elevated levels of maternally transmitted transposon RNAs (Fig 3C). This was also the case of *ago3* single mutant embryos, which displayed nuclear damage (Mani et al, 2014) (Fig S4D) similar to that of *vas$^{D1/D1}$*; *nos-Gal4>GFP-vas$^{WT}$* embryos (Fig 2A and B). In addition to HeT-A RNA, HeT-A/Gag protein was also up-regulated in *vas$^{D1/D1}$*, *mnk$^{P6/P6}$*, and *ago3* embryos during the syncytial blastoderm stage (Fig S4C). At

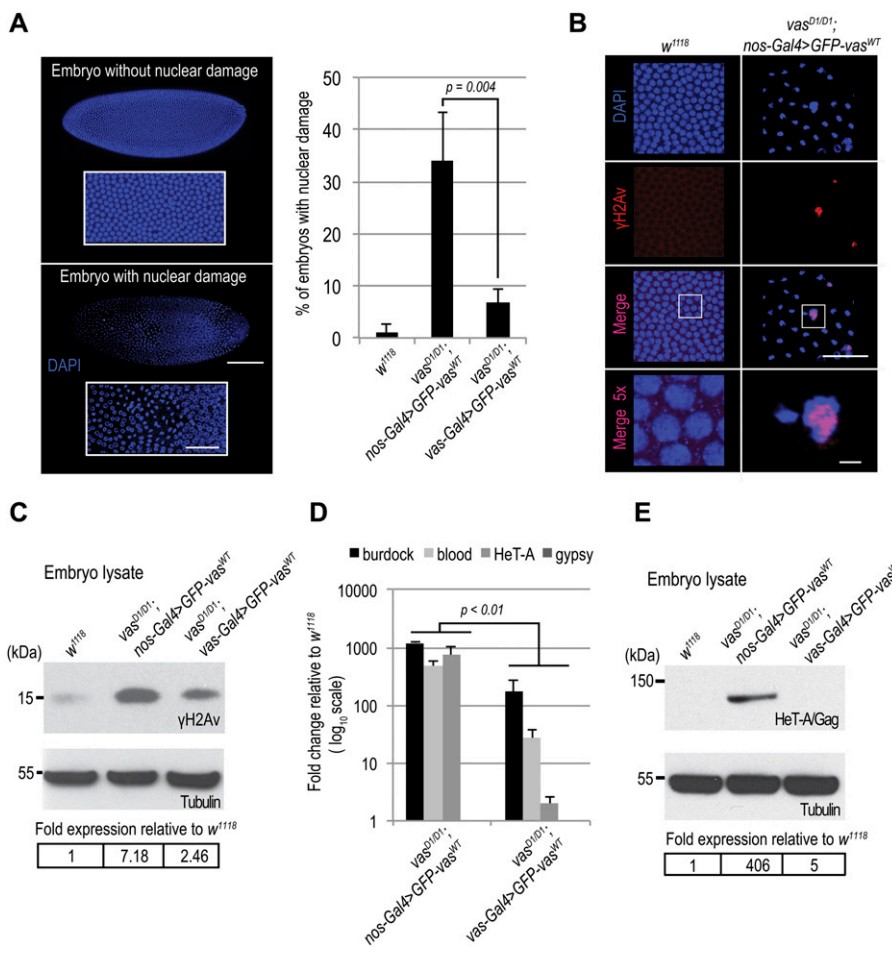

**Figure 2. Maternally transmitted transposon RNAs cause DNA double-strand breaks and nuclear damage in progeny embryos.**
**(A)** Quantification of nuclear damage determined by NucBlue Fixed Cell Stain staining of WT ($w^{1118}$), $vas^{D1/D1}$; nos-Gal4>GFP-vas$^{WT}$, and $vas^{D1/D1}$; vas-Gal4>GFP-vas$^{WT}$ stage 5 embryos. Error bars represent the standard deviation among three biological replicates (Table S4). *P*-value was determined by *t* test. Panel shows an embryo without (top) and an embryo with nuclear damage (bottom). Scale bars indicate 100 μm (embryo) and 10 μm (magnification). **(B)** Immunohistochemical detection of DNA double-strand breaks using antibodies against H2Av pS137 (γH2Av) in WT ($w^{1118}$), and $vas^{D1/D1}$; nos-Gal4>GFP-vas$^{WT}$ stage 5 embryos. Whole embryos are presented in (A). Scale bars indicate 5 and 2 μm (5× magnification). **(C)** Western blot analysis using antibodies against H2Av pS137 (γH2Av) showing protein levels in WT ($w^{1118}$), $vas^{D1/D1}$; nos-Gal4>GFP-vas$^{WT}$, and $vas^{D1/D1}$; vas-Gal4>GFP-vas$^{WT}$ 1- to 3-h-old embryos. Tubulin was used as a loading control. Table shows quantification of γH2Av levels relative to WT. γH2Av signal was normalized to tubulin signal in individual experiments and was set to one in WT. **(D)** qPCR analysis of LTR transposons burdock, blood, and gypsy and non-LTR transposon HeT-A RNAs in $vas^{D1/D1}$; nos-Gal4>GFP-vas$^{WT}$ and $vas^{D1/D1}$; vas-Gal4>GFP-vas$^{WT}$ early embryos. Expression level of transposons in WT ($w^{1118}$) was set to one and normalized to 18S rRNA in individual experiments. Error bars represent the standard deviation among three biological replicates. *t* Test indicated *P*-values for burdock (0.004), blood (0.002), and HeT-A (0.008) lower than 0.01 (indicated in the chart), whereas gypsy levels were not significantly different (*P* = 0.4). **(E)** Western blot analysis using antibodies against HeT-A/Gag showing protein levels in early embryos produced by WT ($w^{1118}$), $vas^{D1/D1}$; nos-Gal4>GFP-vas$^{WT}$, and $vas^{D1/D1}$; vas-Gal4>GFP-vas$^{WT}$ flies. Tubulin was used as a loading control. The table shows quantification of HeT-A/Gag protein levels relative to WT. HeT-A/Gag signal was normalized to tubulin signal in individual experiments and was set to one in WT.

cellularization, the embryos displayed nuclear damage and HeT-A/Gag was present in large foci throughout the embryo (Figs 3D and S4D), resembling the nuclear-damaged $vas^{D1/D1}$; nos-Gal4>GFP-vas$^{WT}$ embryos (Fig 3A, panel b).

We next examined the distribution of HeT-A RNAs and occurrence of DNA DSBs by FISH and antibody staining of γH2Av, respectively. Damaged nuclei in $vas^{D1/D1}$, $mnk^{P6/P6}$ embryos were γH2Av-positive (Figs 4A and S5A), indicating that DNA DSBs cause nuclear defects. HeT-A RNAs localized in large foci in $vas^{D1/D1}$, $mnk^{P6/P6}$ embryos, and was not detectable in WT embryos (Figs 4A and S5A). Although, we did not detect HeT-A transcripts in the damaged nuclei of $vas^{D1/D1}$, $mnk^{P6/P6}$ embryos, the oocyte nucleus was positive both for HeT-A RNA and γH2Av (Fig 4B), indicating the presence of DNA DSBs. Further analysis showed that HeT-A RNA and HeT-A/Gag protein co-localize in the oocyte cytoplasm and nucleus (Fig 5A) indicating that transposon insertions into the maternal genome begin already during oogenesis. Additional FISH analyses showed that in WT egg-chambers, *HeT-A* and *Burdock* transcripts were only detected at sites of transcription in the nurse cell nuclei, whereas in $vas^{D1/D1}$, $mnk^{P6/P6}$, and *ago3* egg-chambers, transcripts of both transposons accumulated within the oocyte along the anterior margin, and around and within the nucleus (Figs 5B and S5B). These results show that in $vas^{D1/D1}$,

$mnk^{P6/P6}$ double and in *ago3* single mutant females up-regulated transposons invade the maternal genome and are transmitted to the progeny, causing severe nuclear defects and embryogenesis arrest. We conclude that tight regulation of transposons throughout oogenesis is essential to maintain genome integrity in the oocyte and in early syncytial embryo, hence for normal embryonic development.

## Discussion

Our study shows that a transient loss of *vas* expression during early oogenesis leads to up-regulation of transposon levels and compromised viability of progeny embryos. The observed embryonic lethality is because of DNA DSBs and nuclear damage that arise as a consequence of the elevated levels of transposon mRNAs and proteins, which are transmitted from the mother to the progeny. We thus demonstrate that transposon silencing in the nurse cells is essential to prevent maternal transmission of transposons and DNA damage, protecting the progeny from harmful transposon-mediated mutagenic effects.

Our finding that suppression of Chk2-mediated DNA damage signaling in loss-of-function *vas* mutant flies restores oogenesis,

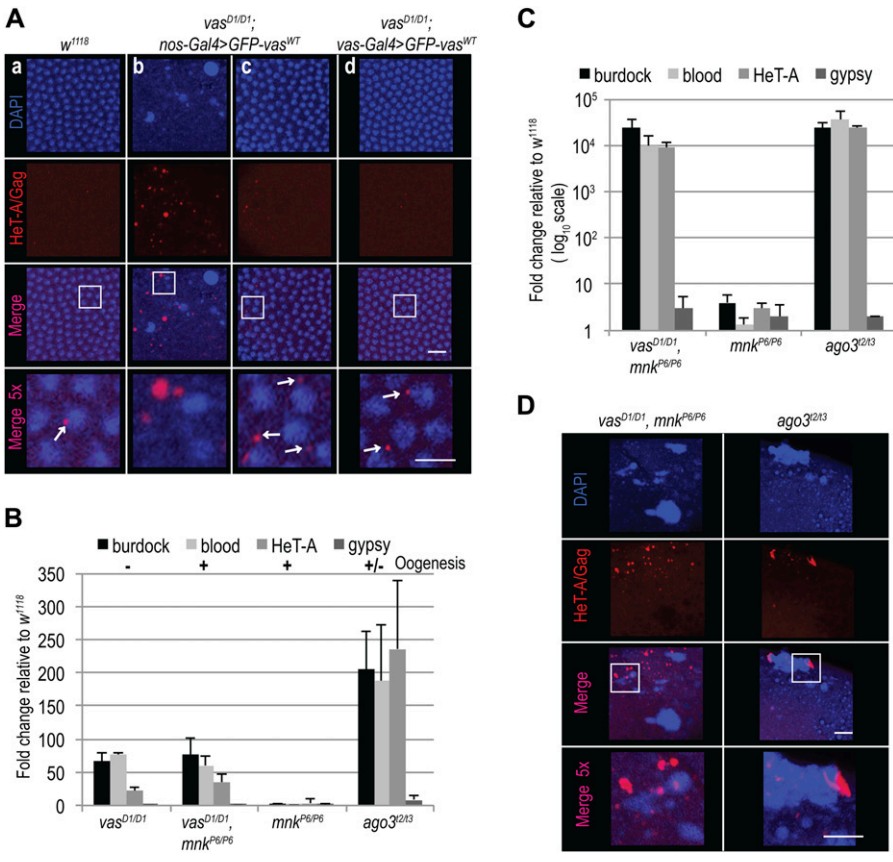

**Figure 3. Loss of Chk2 DNA damage signaling does not restore embryogenesis in *vas* mutant flies.**
**(A)** Immunohistochemical detection of HeT-A/Gag protein in WT (*w*[1118]; a), *vas*[D1/D1]; *nos-Gal4>GFP-vas*[WT] (b and c), and *vas*[D1/D1]; *vas-Gal4>GFP-vas*[WT] (d) stage 5 embryos. Arrows indicate WT localization of HeT-A/Gag. Staining of the whole embryos is presented in Fig S3A. Scale bars indicate 10 and 5 µm (5× magnification). **(B)** qPCR analysis of LTR transposons *burdock*, *blood*, and *gypsy* and non-LTR transposon *HeT-A* RNAs in ovaries from *vas*[D1/D1] single and *vas*[D1/D1], *mnk*[P6/P6] double mutant flies, and *mnk*[P6/P6] and *ago3*[t2/t3] mutant flies. The expression level of transposons in WT (*w*[1118]) was set to one and normalized to rp49 mRNA in individual experiments. Error bars represent the standard deviation among three biological replicates. Oogenesis completion is indicated with + and –. **(C)** qPCR analysis of LTR transposons *burdock*, *blood*, and *gypsy* and non-LTR transposon *HeT-A* RNAs in early embryos produced by *vas*[D1/D1], *mnk*[P6/P6] double mutant, and *mnk*[P6/P6] and *ago3*[t2/t3] single mutant flies. The expression level of transposons in WT (*w*[1118]) was set to one and normalized to 18S rRNA in individual experiments. Error bars represent the SD among three biological replicates. **(D)** Immunohistochemical detection of HeT-A/Gag protein in stage 5 embryos from *vas*[D1/D1], *mnk*[P6/P6] double mutant and *ago3*[t2/t3] single mutant flies. Staining of the whole embryos is presented in Fig S4D. Scale bars indicate 10 and 5 µm (5× magnification).

and egg production demonstrates that Chk2 is epistatic to *vas*. However, hatching is severely impaired, because of the DNA damage sustained by the embryos. The defects displayed by *vas*, *mnk* double mutant embryos resembled those of PIWI (*piwi*, *aub*, and *ago3*) single and *mnk*; PIWI double mutant embryos (Klattenhoff et al, 2007; Mani et al, 2014). Earlier observation that inactivation of DNA damage signaling does not rescue the development of PIWI mutant embryos led to the assumption that PIWI proteins might have an essential role in early somatic development, independent of cell cycle checkpoint signaling (Mani et al, 2014). By tracing transposon protein and RNA levels and localization from the mother to the early embryos, we suggest that, independent of Chk2 signaling, de-repressed transposons are responsible for nuclear damage and embryonic lethality. Our study indicates that transposon insertions occur in the maternal genome where they cause DNA DSBs that together with transposon RNAs and proteins are passed on to the progeny embryos. Transposon activity and consequent DNA damage in the early syncytial embryo cause aberrant chromosome segregation, resulting in unequal distribution of the genetic material, nuclear damage and ultimately embryonic lethality. Our study shows that early *Drosophila* embryos are defenseless against transposons and will succumb to their mobilization if the first line of protection against selfish genetic elements in the nuage of nurse cell fails.

A recent study showed that in *p53* mutants, transposon RNAs are up-regulated and accumulate at the posterior pole of the oocyte, without deleterious effects on oogenesis or embryogenesis (Tiwari et al, 2017). It is possible that the absence of pole plasm in *vas* mutants (Lehmann & Ephrussi, 1994) results in the release of the transposon products and their ectopic accumulation in the oocyte. Localization of transposons to the germ plasm (Tiwari et al, 2017) may restrict their activity to the future germline and protect the embryo soma from transposon activity. Transposon-mediated mutagenesis in the germline would produce genetic variability, a phenomenon thought to play a role in the environmental adaptation and evolution of species. It would therefore be of interest to determine the role of pole plasm in transposon control in the future.

Transposon up-regulation in the *Drosophila* female germline triggers a DNA damage-signaling cascade (Chen et al, 2007; Klattenhoff et al, 2007). In *aub* mutants, before their oogenesis arrest occurs, Chk2-mediated signaling leads to phosphorylation of Vasa, leading to impaired *grk* mRNA translation and embryonic axis specification (Klattenhoff et al, 2007). Considering the genetic interaction of *vas* and *mnk* (Chk2) and the fact that Vasa is phosphorylated in Chk2-dependent manner (Abdu et al, 2002; Klattenhoff et al, 2007), it is tempting to speculate that phosphorylation of Vasa might stimulate piRNA biogenesis, reinforcing transposon silencing and thus minimizing transposon-induced DNA damage (Fig 5C). The arrest of embryonic development as a first, and arrest of oogenesis as an ultimate response to DNA damage, thus, prevents the spreading of detrimental transposon-induced mutations to the next generation.

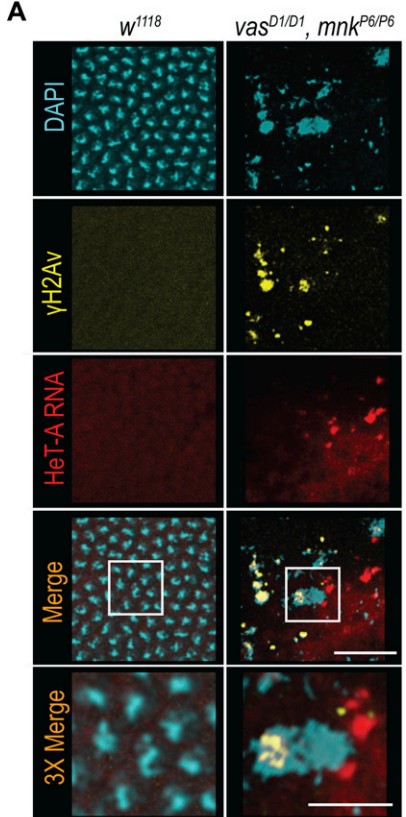

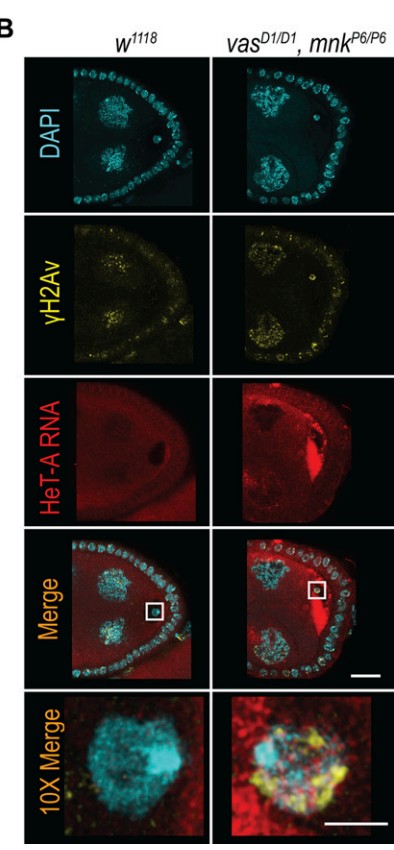

**Figure 4. Transposons invade maternal genome and cause DNA DSBs in vas, mnk double mutant flies. (A, B)** In situ detection of HeT-A mRNA by FISH and immunohistochemical detection of DNA DSBs using antibodies against H2Av pS137 (γH2Av) in WT (*w^1118*) and *vas^D1/D1*, *mnk^P6/P6* double mutant embryos (A) and ovaries (B). Scale bars in (A) indicate 5 and 2 μm (3× magnification); scale bars in (B) indicate 20 and 5 μm (10× magnification).

## Experimental procedures

### Fly stocks and husbandry

The following *Drosophila* stocks were used: *w^1118*; *b^1*, *vas^D1*/CyO (*vas^3*, Tearle and Nusslein-Volhard, 1987; Lasko & Ashburner, 1990); *b^1*, *vas^Q7*, *pr^1*/CyO (*vas^7*, Tearle and Nusslein-Volhard, 1987; Lasko & Ashburner, 1990); *vas^D1*/CyO; *nos-Gal4-VP16*/TM2 (Xiol et al, 2014); *vas-Gal4* (gift of Jean-René Huynh); *GFP-vas^WT*/TM2 (Xiol et al, 2014); *mnk^P6*/CyO (Brodsky et al, 2004); *bw^1*; *st^1*, *ago3^t2*/TM6B, Tb^+ (FBst0028269), *bw^1*; *st^1*; *ago3^t3*/TM6B, Tb^1 (FBst0028270). All flies were kept at 25°C on standard *Drosophila* medium.

### Generation of mnk, vas double mutant flies

To generate *mnk*, *vas* double mutants, +, +, *mnk^P6 [P{lacW}]*/CyO and *b^1*, *vas^D1*, +/CyO flies were crossed. F1 progeny +, +, *mnk^P6 [P{lacW}]*/*b^1*, *vas^D1*, + females were then crossed to males of the balancer stock *CyO/if*. F2 progeny were screened for red eyes (*mnk^P6* marker *P{lacW}*) and 200 individual red-eyed flies were mated to CyO/if balancer flies. F3 generation stocks were established and screened for non-balanced flies of a dark body color (homozygous for *b^1*, a marker of the original *vas^D1* chromosome). Three lines were obtained and tested for presence of the *vas^D1* mutation by Western blotting (Fig S2C) and for presence of the *mnk* mutation by RT–PCR (Fig S3B). A scheme of the crosses and recombination is shown in Table S6 and sequences of primers used for RT–PCR reaction are shown in Table S7.

### Fecundity and hatching assays

Virgin females of all genetic backgrounds tested were mated with *w^1118* males for 24 h at 25°C. The crosses were then transferred to apple-juice agar plates, and eggs collected in 24 h intervals over 3–4 d. The number of eggs laid on each plate was counted; the plates were kept at 25°C for 2 d, then the number of hatched larvae counted. Experiments were performed in three independent replicates represented in Tables S2, S3, and S5. *w^1118* females were used as a control.

### Ovarian morphology and Vasa localization analysis

Ovaries of 3- to 7-d old flies were dissected in PBS. Ovarian morphology was evaluated under an Olympus SZX16 stereo microscope. Vasa localization was assessed in ovaries of 3- to 7-d old flies expressing the GFP-Vasa proteins after fixation in 2% PFA and 0.01% Triton X-100 for 15 min at RT. Fixed ovaries were mounted on glass slides and GFP fluorescence examined under a Zeiss LSM 780 confocal microscope. Vasa localization in WT and *vas* mutant ovaries and progeny embryos was analyzed by antibody staining (see below). Nuclei were visualized with NucBlue Fixed Cell Stain (Thermo Fisher Scientific).

### Immunohistochemical staining of ovaries and embryos

Freshly hatched females were mated with WT males and kept for 2–3 d on yeast at 25°C before dissection. Ovaries were dissected in PBS and immediately fixed by incubation at 92°C for 5 min in preheated fixation buffer (0.4% NaCl, 0.3% Triton X-100 in PBS),

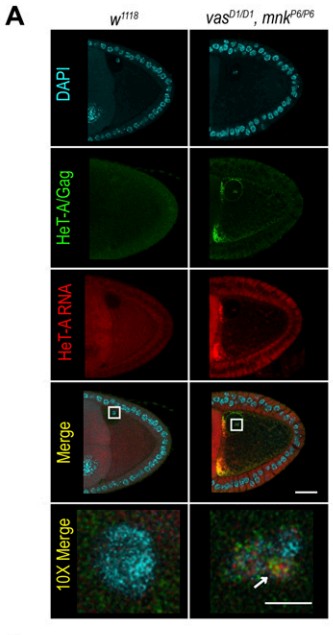

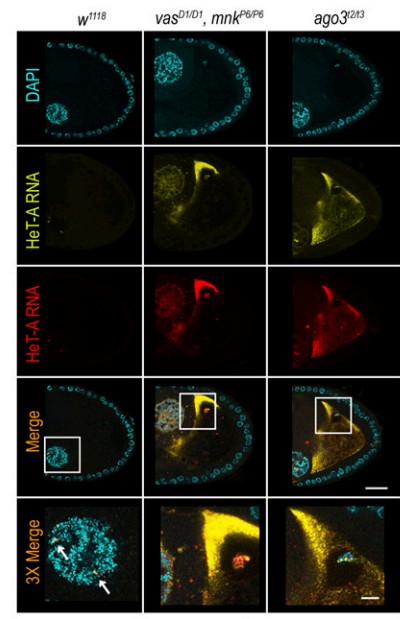

**Figure 5. Vasa couples the DNA damage response machinery and the piRNA pathway in Drosophila female germline.**
**(A)** In situ detection of HeT-A mRNA by FISH and immunohistochemical detection of HeT-A/Gag protein in WT ($w^{1118}$) and $vas^{D1/D1}$, $mnk^{P6/P6}$ double mutant ovaries. Arrow indicates co-localization of HeT-A mRNA and HeT-A/Gag protein signals. Scale bars indicate 20 and 5 $\mu m$ (10× magnification). **(B)** In situ detection of HeT-A mRNA by FISH in WT ($w^{1118}$), $vas^{D1/D1}$, $mnk^{P6/P6}$ double mutant, and $ago3^{t2/t3}$ single mutant ovaries. Arrows indicate sites of HeT-A mRNA transcription. Scale bars indicate 20 and 5 $\mu m$ (3× magnification). **(C)** In WT flies, the occurrence of DNA DSBs activates Chk2 kinase that regulates several mechanisms that together antagonize deleterious effects of DNA damage. Chk2 might directly or indirectly target Vasa that in turn affects piRNA biogenesis and transposon control, reducing the transposon-induced DSBs. Accordingly, DNA damage induced by high levels of transposons in *vas* mutants triggers DNA damage–induced apoptosis resulting in oogenesis arrest. Oogenesis can be restored by depletion of Chk2; however, transposon deregulation persists and causes severe nuclear damage and embryogenesis arrest preventing distribution of transposon-induced, detrimental mutations within the population.

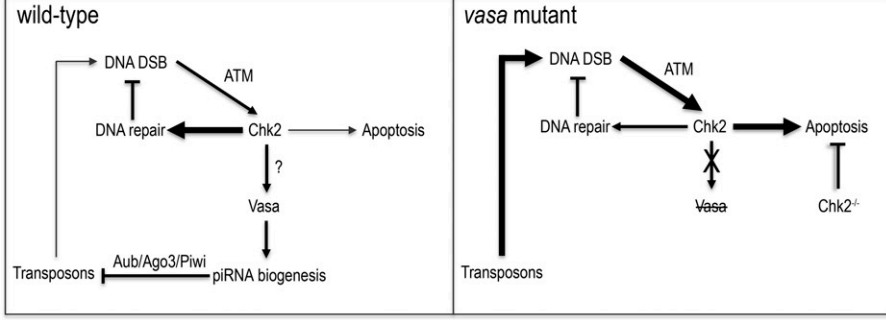

followed by extraction in 1% Triton X-100 for 1 h at RT. Fixed ovaries were incubated with primary antibodies against Vasa (rat; 1:500; Tomancak et al, 1998) or HeT-A/Gag (rabbit 1:100; gift of Elena Casacuberta). The secondary antibodies were used: Alexa 488 conjugated goat anti-rabbit (1:1,000; Invitrogen) and Alexa 647 conjugated donkey anti-rat IgG (1:1,000; Jackson ImmunoResearch). Nuclei were stained with NucBlue Fixed Cell Stain (Thermo Fisher Scientific).

For embryo staining, freshly hatched females were mated with WT males and fed with yeast for 2–3 d at 25°C before egg collection. Embryos (0–1 h or 1–3 h) were collected and dechorionated in 50% bleach, then fixed by incubation at 92°C for 30 s in preheated fixation buffer (0.4% NaCl, 0.3% Triton X-100 in PBS), followed by devitellinization by rigorous shaking in a 1:1 mix of heptane and methanol. After washing in 0.1% Tween-20, embryos were either immediately incubated with primary antibodies against Vasa (rat; 1:500; Tomancak et al, 1998) or HeT-A/Gag (rabbit 1:100; gift from Elena Casacuberta), or stored in methanol at –20°C for staining later on. For detection of DSBs, embryos (1–3 h) were collected and dechorionated in 50% bleach, fixed for 25 min at RT in the heptane/4% formaldehyde interface, and devitellinized by rigorous shaking after adding 1 V of methanol. After washing in 0.1% Tween-20, the embryos were either immediately incubated with

primary antibodies against H2Av pS137 (γH2Av; rabbit; 1:5,000; Rockland) or stored in methanol at 20°C for later staining. The following secondary antibodies were used: Alexa 488 conjugated goat anti-rabbit (1:1,000; Invitrogen), Alexa 647 conjugated donkey anti-rat IgG (1:1,000; Jackson ImmunoResearch), and Alexa 647 conjugated goat anti-rabbit IgG (1:1,000; Invitrogen). Nuclei were stained with NucBlue Fixed Cell Stain (Thermo Fisher Scientific). The samples were observed using a Zeiss LSM 780 or Leica SP8 confocal microscope.

### Fluorescent in situ RNA hybridization

All FISH experiments were performed as described in Gáspár et al (2018). In brief, ovaries were dissected in PBS and immediately fixed in 2% PFA, 0.05% Triton X-100 in PBS for 20 min at RT. Embryos (1–3 h) were collected and dechorionated in 50% bleach, fixed for 25 min at RT in the heptane/2% PFA interface and devitellinized by vigorous shaking after adding 1 V of methanol. After washing in PBT (phosphate buffered saline + 0.1% Triton X-100), samples were treated with 2 $\mu g/ml$ proteinase K in PBT for 5 min and then were subjected to 95°C in PBS + 0.05% SDS for 5 min. Proteinase K treatment was omitted when samples were subsequently to be immunohistochemically stained (see below). Samples were pre-hybridized in 200 $\mu l$ hybridization

buffer (300 mM NaCl, 30 mM sodium citrate pH 7.0, 15% ethylene carbonate, 1 mM EDTA, 50 μg/ml heparin, 100 μg/ml salmon sperm DNA, and 1% Triton X-100) for 10 min at 42°C. Fluorescently labeled oligonucleotides (12.5–25 nM) were pre-warmed in hybridization buffer and added to the samples. Hybridization was allowed to proceed for 2 h at 42°C. Samples were washed 3 times for 10 min at 42°C in pre-warmed buffers (1× hybridization buffer, then 1× hybridization buffer:PBT 1:1 mixture, and then 1× PBT). The final washing step was performed in pre-warmed PBT at RT for 10 min. The samples were mounted in 80% 2,2-thiodiethanol in PBS and analyzed on a Leica SP8 confocal microscope.

For simultaneous FISH and immunohistochemical staining, ovaries and embryos were fixed as described above. Samples were simultaneously incubated with fluorescently labeled oligonucleotides (12.5–25 nM) complementary to HeT-A RNA and primary antibodies against γH2Av (rabbit; 1:5,000; Rockland) or HeT-A/Gag (rabbit 1:100; gift of Elena Casacuberta) overnight at 28°C in PBT. Samples were washed 2 times for 20 min at 28°C in PBT and subsequently incubated with secondary Alexa 488 conjugated goat anti-rabbit antibodies (1:1,000; Invitrogen). The samples were mounted in 80% 2,2-thiodiethanol in PBS and analyzed on a Leica SP8 confocal microscope.

### Labeling of DNA oligonucleotides for fluorescent in situ RNA hybridization

Labeling of the oligonucleotides was performed as described in Gáspár et al (2018). Briefly, non-overlapping arrays of 18–22 nt long DNA oligonucleotides complementary to *HeT-A* or *Burdock* RNA (Table S7) were selected using the smFISHprobe_finder.R script (Gáspár et al, 2018). An equimolar mixture of oligos for a given RNA was fluorescently labeled with Alexa 565- or Alexa 633-labeled dideoxy-UTP using terminal deoxynucleotidyl transferase. After ethanol precipitation and washing with 80% ethanol, fluorescently labeled oligonucleotides were reconstituted with nuclease-free water.

### Protein extraction and Western blotting

To generate ovarian lysates, around 20 pairs of ovaries from 3- to 7-d-old flies were homogenized in protein extraction buffer (25 mM Tris pH 8.0, 27.5 mM NaCl, 20 mM KCl, 25 mM sucrose, 10 mM EDTA, 10 mM EGTA, 1 mM DTT, 10% [vol/vol] glycerol, 0.5% NP40, 1% Triton X-100, and 1× Protease inhibitor cocktail [Roche]). For embryo lysates, 0- to 1-h-old or 1- to 3-h-old embryos were collected from apple-juice agar plates and homogenized in protein extraction buffer. Samples were incubated on ice for 10 min, followed by two centrifugations, each 15 min at 16,000 *g*. 50–100 μg of total protein extracts were solubilized in SDS sample buffer by boiling at 95°C for 5 min, then analyzed by SDS polyacrylamide gel electrophoresis (4–12% NuPAGE gel; Invitrogen). Western blotting was performed using antibodies against Vasa (rat; 1:3,000; Tomancak et al [1998]), HeT-A/Gag (rabbit 1:750; gift from Elena Casacuberta), H2Av pS137 (γH2Av; rabbit; 1:1,000; Rockland), and Tub (mouse; 1:10,000; T5168; Sigma-Aldrich). Western blot analyses were performed in duplicates.

Quantification of relative protein expression levels was performed using ImageJ. A frame was placed around the most prominent band on the image and used as a reference to measure the mean gray value of all other protein bands and the background. Next, the inverted value of the pixel density was calculated for all measurements by deducting the measured value from the maximal pixel value. The net value of target proteins and the loading control was calculated by deducting the inverted background from the inverted protein value. The ratio of the net value of the target protein and the corresponding loading control represents the relative expression level of the target protein. Fold-change was calculated as the ratio of the relative expression level of the target protein in the WT control over that of a specific sample.

### RNA extraction and quantitative PCR analysis

Total RNA was extracted from ovaries of 3- to 7-d-old flies or 0- to 1-h-old embryos using Trizol reagent (Thermo Fisher Scientific). For first-strand cDNA synthesis, RNA was reverse-transcribed using a QuantiTect Reverse Transcription Kit (QIAGEN). Quantitative PCR (qPCR) was performed on a StepOne real-time PCR system (Thermo Fisher Scientific) using SYBR Green PCR Master Mix (Thermo Fisher Scientific). Relative RNA levels were calculated by the $2^{-\Delta\Delta CT}$ method (Livak & Schmittgen, 2001) and normalized to rp49 mRNA levels for ovaries, and 18S rRNA for embryos. Fold-enrichments were calculated by comparison with the respective RNA levels in $w^{1118}$ flies. Sequences of primers used for qPCR reaction are shown in Table S7.

## Data Availability

The authors declare that all data supporting the findings of this study are available within the manuscript and its supplementary files.

## Supplementary Information

## Acknowledgments

We thank Elena Casacuberta for the gift of antibodies against HeT-A/Gag, and Beat Suter and Jean-René Huynh for fly stocks. We are grateful to Anna Cyrklaff and Alessandra Reversi for their help with experiments. We thank the EMBL Advanced Light Microscopy Core Facility for use of its microscopes. This work was funded by the European Molecular Biology Laboratory (EMBL) and Z Durdevic by a postdoctoral fellowship from the EMBL Interdisciplinary Postdoc Program under Marie Curie COFUND actions.

### Author Contributions

A Ephrussi: conceptualization, resources, formal analysis, supervision, funding acquisition, methodology, project administration, and writing—original draft, review, and editing.
Z Durdevic: conceptualization, formal analysis, funding acquisition, validation, investigation, visualization, methodology, project administration, and writing—original draft, review, and editing.
RS Pillai: conceptualization, supervision, and writing—original draft, review, and editing.

### Conflict of Interest Statement

The authors declare that they have no conflict of interest.

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
