## [Reviewer comments · Life Science Alliance]

Transposon silencing in the *Drosophila* female germline is essential for genome stability in progeny embryos

Zeljko Durdevic, Ramesh S. Pillai and Anne Ephrussi

DOI: 10.26508/lsa.201800179

Review timeline:

Submission Date:	27 August 2018
Editorial Decision:	27 August 2018
Revision Received:	31 August 2018
Editorial Decision:	3 September 2018
Accepted:	5 September 2018

Report:

(Note: Letters and reports are not edited. The original formatting of letters and referee reports may not be reflected in this compilation.)

Please note that the manuscript was previously reviewed at another journal and the reports were taken into account in inviting a revision for publication at *Life Science Alliance* prior to submission to *Life Science Alliance*.

1st Editorial Decision

27 August 2018

Thank you for transferring your manuscript entitled "Transposon silencing in the *Drosophila* female germline ensures genome stability in progeny embryos" to Life Science Alliance. The manuscript was assessed by expert reviewers at another journal before, and the journal editors have transferred those reports to us with your permission.

The reviewers who assessed your work at the other journal noted that your observations add to our understanding of the role of Vasa in transposon silencing and oogenesis, but they would have expected further reaching insight. This is not a concern for publication in Life Science Alliance. The reviewers furthermore pointed out that a few controls were missing, and that more quantifications, clarifications and information should be provided. As pre-discussed with you prior to submission to our journal, we would like to invite you to address these latter points and to submit a revised version of your manuscript for publication in Life Science Alliance.

Thank you for this interesting contribution to Life Science Alliance. We are looking forward to receiving your revised manuscript.

REFeree REPORTS OBTAINED DURING PEER REVIEW ELSEWHERE

Referee #1:

In this study Durdevic et.al., have elucidated the role of Vasa in regulating transposons during early embryo development. By utilizing two GAL4 lines, nos and vas, they have shown that even transient absence of vasa during oogenesis results in an increased expression of transposons and viability of progeny. Additionally, increased transposon levels in nosGAL4 driven rescue compared to vasGAL4 driven rescue does not perturb progression of oogenesis but rather embryo hatching, indicating that oogenesis is not affected below a threshold of transposon activity. The authors show that low viability of vas mutant embryos is a result of DNA damage which is a result of high levels of maternally transmitted transposons. Moreover, the authors show that removal of DNA damage signaling pathway (Chk2) does rescue germline defects observed in vas mutants but does not de-repress transposon levels. This observation shows that germline defects observed in vas mutants results from the activation of the Chk2 pathway. Next, the authors show that similar to ago3

mutants, *vas* and *mnk* double mutant embryos show an upregulation of transposons (HeT-A) and DNA damage. Lastly, the authors provide evidence via FISH experiments that HeT-A RNA and HeT-A/Gag protein colocalized with the oocyte nucleus in *vas*, *mnk* and *ago3* mutant egg chambers but was not detectable in wild type oocyte. Altogether, the authors have shown that in absence of *vas*, *mnk* and *ago3* transposons invade the maternal genome during oogenesis causing nuclear defects and embryogenesis arrest.

This study is of potential interest and identifies a key regulator of transposon regulation during *Drosophila* early embryo development. However, several concerns, listed below, including analysis of experimental data needs to be addressed. Moreover, the lack of mechanistic evidence of how *vas* inhibits transposon inheritance during oogenesis independent of the Chk2 pathway weaken my enthusiasm for publication for now.

Concerns:

1. Fig. 2E, S2C, S4C: The authors need to mention how many trials were performed in all the Western Blot analysis and if the results were significant.
2. Fig. 2B: The authors mention that γ H2Av are higher in *vas*; *nosGAL4*>GFP-VasWT when compared to *vas*; *vasGAL4*>GFP-VasWT via Western analysis, but how many embryos do they observe that have nuclear damage in these genotypes? Is it significantly different?
3. Fig. 2C: The authors report an upregulation of γ H2Av protein levels in both *vas*; *nosGAL4*>GFP-VasWT and *vas*; *vasGAL4*>GFP-VasWT compared to wild type. Do the authors observe absolutely no damaged nuclei in *vas*; *vasGAL4*>GFP-VasWT embryos, even when there is higher amount of γ H2Av protein present? How do the authors account for this difference?
4. The authors show that HeT-A/Gag protein accumulated in large foci in nuclear damaged *vas*; *nosGAL4*>GFP-VasWT embryos (Fig. 3A). This supports the high level of HeT-A RNA quantified by qPCR (Fig. 2D) However, in page 8, they mention that no HeT-A RNA (via FISH) was detected in damaged nuclei of *vas*,*mnk* double mutants, but HeT-A RNA levels and protein levels were high in these embryos (Fig. 3C, S4C and D). How do the authors explain this discrepancy? Do they detect HeT-A RNA (via FISH) in nuclear damaged embryos of *vas*; *nosGAL4*>GFP-VasWT? What does Burdock RNA (via FISH) look like in these DNA damaged embryos?
Note: In page 6 and 7 line 4, did the authors mean to refer to Supplementary Figure S3A as there are no S3B with panels a-d?
5. The authors should also provide FISH analysis of HeT-A and Burdock RNA in *vas*; *nosGAL4*>GFP-VasWT and *vas*; *vasGAL4*>GFP-VasWT ovaries. Do they have the same pattern seen in *vas*,*mnk* double mutants? Theoretically there should be no HeT-A RNA in the oocyte of *vas*; *vasGAL4*>GFP-VasWT as they have viable embryos.
6. The authors should show the localization and expression pattern of HeT-A RNA and HeT-A/Gag protein in ovarioles of *nosGAL4* and *VasGAL4* ovarioles, especially between stages 2 and 6 to show how early are these transposons invade the oocyte and transient loss of *vas* is sufficient for this inheritance.

Minor points:

1. The authors should not say nanos promoter was inactive between stages 2 and 6 as they have not shown this. A better word would be "attenuated".

Referee #2:

The manuscript by Durdevic et al. reports two interesting observations: (1) Transient loss of *Vasa* expression during early oogenesis (stages 2-6) leads to transposon up-regulation in the germline, which does not affect the progression of oogenesis but leads to increased DNA damage in egg chambers and resulting early embryos. (2) A Chk2 mutation restores oogenesis of *Vasa* mutant females without silencing transposons. These observations add to our understanding of the role of *Vasa* in transposon silencing and oogenesis. However, given the known role of *Vasa* in the piRNA

pathway and the known role of the piRNA pathway in oogenesis (e.g. Klattenhoff et al. 2007), the findings reported in this manuscript are largely confirmatory. They bear limited biological novelty. In addition, the manuscript lacks molecular/mechanistic insights to key issues such as the nature of Ch2 epistasis over vasa. For example, the author did not even attempt to examine whether the Chk2 mutation changes the phosphorylation of Vasa as previously reported by Klattenhoff et al. (2007). Lastly, some results are weak and some interpretations are either biased or over-stated (see below). Therefore, the manuscript not suitably for publication in EMBO Report at this stage.

Specific comments:

(1) Fig 1A and p4, para 2, the author stated that "when driven by nos-Gal4, GFP-VasWT had no effect on transposon levels", and they attributed this to "transposon de-repression that lack of Vasa between stages 2 and 6 of oogenesis". It is difficult to rationalize why stages 2-6 are so crucial for transposon silencing. Did the authors make sure that the qPCR is completely linear to the RNA levels? A nonlinear qPCR result (which often is the case unless the reaction is well-calibrate) may preferentially reduce the difference between the levels of two RNA samples that are both well above the detection limit. This could be the case for the vas mutant ovaries and the nos-Gal4>GFP-vasa ovaries. In support of this possibility, the transgenic ovaries did display reduced transposon RNA levels (unlike the authors' claimed no difference), even though the reduction may not be statistically significant (?).

(2) Fig 1A: It seems that Vasa is more effective in silencing Het-A (non-LTR) transposons than LTR transposons. The authors need to point out this and ideally also provide an explanation.

(3) P5, para 1: "The fact that in spite of transposon up-regulation oogenesis and egg-laying rates were fully restored in vasD1/D1 flies (Figure 1A, indicated by + and - and Supplementary Figure S1D-F) indicates that oogenetic processes are not affected below a certain threshold of transposon activity." These statements are inaccurate and somewhat misleading. First of all, the egg laying rate is not fully, but only largely, restored, with nos-Gal4>GFP-vasa females showing less restoration than vas-Gal4>GFP-vasa females. This is consistent with the notion that transposon activation affects oogenesis but does not completely block it, unless the level of activation is extremely high to cause ovarian dysgenesis.

BTW, "oogenetic" should be "oogenic".

(4) Do vasD1/D1 females lay eggs at all? This needs to be reported and included in Fig S1E, even if the mutant females do not lay eggs. If eggs are laid by the mutant, Fig 2E need to include a 0-1 hour vasa mutant embryo control to compare the levels of HetA/Gag protein in these embryos with those in nos-Gal4>GFP-vasa and vas-Gal4>GFP-vasa embryos. Otherwise, it is difficult to make a proper conclusion from the western blot. In addition, the protein loading of different lanes needs to be more uniform to allow more meaningful comparison. Likewise, Fig3A and other relevant figures also need to include a control of 0-1 hour vasa mutant embryos.

(5) P7, line 4, "Supplementary Figure S3B panel b" should be "Supplementary Figure S3A panel b".

(6) P9, Discussion: "we have shown that, independent of Chk2 signaling, de-repressed transposons are responsible for nuclear damage and embryonic lethality." This is an over-statement. The authors established a solid correlation, but have not definitively shown, that the de-repressed transposons are responsible for nuclear damage and embryonic lethality.

Referee #3:

The authors study the effects of vasa mutations on oogenesis as well as embryogenesis of *Drosophila*. The authors find that depletion of the fly Chk2 ortholog mnk restores oogenesis in vas mutants, but does not suppress defects in transposon silencing or DSB-induced nuclear damage and embryonic lethality. This paper suggests that the Vasa-dependent protection against transposons in the nurse cells is essential to prevent DNA damage-induced arrest of embryonic development. The novelty of the present study however has been dampened by previous studies by others demonstrating that depletion of the mnk restores oogenesis in mutations in piRNA pathway components (Chen et al., 2007; Klattenhoff et al., 2007; Pane et al., 2007; and others). In addition,

Haifan Lin and colleagues made double mutants of *mnk* and *ago3*, a component of the piRNA pathway, to examine the role of DNA damage signaling in Ago3-depleted embryo defects and found that while egg laying was rescued in the double mutants, egg hatching remained defective (see Figure 6 in the paper of Mani et al., 2014). Overall, with a few relatively small exceptions, this paper mostly repeats data already published by other groups.

Other comments:

1. Only Figure 1A and Figure 3B show the results with *vas*[D1/D1]. The authors should show the results with *vas*[D1/D1] in all other experiments (in particular, Fig. S1E, Fig. 1C & D, Fig. S3C, and Fig S4A) as controls.

2. Figure 2A: It is unclear how the authors define "nuclear damage."

1st Revision – authors' response

31 August 2018

Referee 1

In this study Durdevic et al., have elucidated the role of Vasa in regulating transposons during early embryo development. By utilizing two GAL4 lines, *nos* and *vas*, they have shown that even transient absence of vasa during oogenesis results in an increased expression of transposons and viability of progeny. Additionally, increased transposon levels in *nos*GAL4 driven rescue compared to *vas*GAL4 driven rescue does not perturb progression of oogenesis but rather embryo hatching, indicating that oogenesis is not affected below a threshold of transposon activity. The authors show that low viability of *vas* mutant embryos is a result of DNA damage which is a result of high levels of maternally transmitted transposons. Moreover, the authors show that removal of DNA damage signaling pathway (*Chk2*) does rescue germline defects observed in *vas* mutants but does not de-repress transposon levels. This observation shows that germline defects observed in *vas* mutants results from the activation of the *Chk2* pathway. Next, the authors show that similar to *ago3* mutants, *vas* and *mnk* double mutant embryos show an upregulation of transposons (*HeT-A*) and DNA damage. Lastly, the authors provide evidence via FISH experiments that *HeT-A* RNA and *HeT-A/Gag* protein colocalized with the oocyte nucleus in *vas*, *mnk* and *ago3* mutant egg chambers but was not detectable in wild type oocyte. Altogether, the authors have shown that in absence of *vas*, *mnk* and *ago3* transposons invade the maternal genome during oogenesis causing nuclear defects and embryogenesis arrest.

This study is of potential interest and identifies a key regulator of transposon regulation during *Drosophila* early embryo development. However, several concerns, listed below, including analysis of experimental data needs to be addressed. Moreover, the lack of mechanistic evidence of how *vas* inhibits transposon inheritance during oogenesis independent of the *Chk2* pathway weaken my enthusiasm for publication for now.

Concerns:

1. Fig. 2E, S2C, S4C: The authors need to mention how many trials were performed in all the Western Blot analysis and if the results were significant. [MO: add]

We now specify the Experimental Procedures (Lines 401-402) that the western blots were performed in duplicate.

2. Fig.2B: The authors mention that γ H2Av are higher in *vas*; *nos*GAL4>*GFP-Vas*WT when compared to *vas*; *vas*GAL4>*GFP-Vas*WT via Western analysis, but how many embryos do they observe that have nuclear damage in these genotypes? Is it significantly different? [MO: add]

We observe ca. 7% of *vas*D1/D1; *vas*GAL4>*GFP-Vas*WT, and ca. 34% of *vas*D1/D1; *nos*GAL4>*GFP-Vas*WT embryos that have nuclear damage (p=0.004). These data are presented in Figure 2A and the numbers of embryos counted are indicated in Supplementary Table S5 (as mentioned in the figure legend).

3. Fig. 2C: The authors report an upregulation of γ H2Av protein levels in both *vas*; *nos*GAL4>*GFP-Vas*WT and *vas*; *vas*GAL4>*GFP-Vas*WT compared to wild type. Do the authors observe absolutely no damaged nuclei in *vas*; *vas*GAL4>*GFP-Vas*WT embryos, even when there is

higher amount of γ H2Av protein present? How do the authors account for this difference? [MO: discuss or rebuttal]

As indicated above (point 2), the *vasD1/D1; vasGAL4>GFP-VasWT* embryos show nuclear damage, consistent with the elevated levels of γ H2Av. This is shown in Figure 2A and Supplementary Table S5.

4. The authors show that HeT-A/Gag protein accumulated in large foci in nuclear damaged *vas; nosGAL4>GFP-VasWT* embryos (Fig. 3A). This supports the high level of HeT-A RNA quantified by qPCR (Fig. 2D) However, in page 8, they mention that no HeT-A RNA (via FISH) was detected in damaged nuclei of *vas,mnk* double mutants, but HeT-A RNA levels and protein levels were high in these embryos (Fig.3C, S4C and D). How do the authors explain this discrepancy? Do they detect HeT-A RNA (via FISH) in nuclear damaged embryos of *vas; nosGAL4>GFP-VasWT*? What does Burdock RNA (via FISH) look like in these DNA damaged embryos? [MO: discuss or rebuttal]

It is correct that by FISH we detect no *HeT-A* RNA in the nuclei of *vas,mnk* double mutant embryos. However, we do detect *HeT-A* RNA in large (non-nuclear) foci that are randomly distributed throughout those embryos (please see Figure 4A and Supplementary Figure S5A). Hence there is no discrepancy between the qPCR and western blot analyses and the FISH analysis.

Note: In page 6 and 7 line 4, did the authors mean to refer to Supplementary Figure S3A as there are no S3B with panels a-d? [MO: fix]

We apologize for the confusion. We have corrected this in the revised manuscript (Lines 181-186).

5. The authors should also provide FISH analysis of HeT-A and Burdock RNA in *vas; nosGAL4>GFP-VasWT* and *vas; vasGAL4>GFP-VasWT* ovaries. Do they have the same pattern seen in *vas,mnk* double mutants? Theoretically there should be no HeT-A RNA in the oocyte of *vas; vasGAL4>GFP-VasWT* as they have viable embryos. [MO: do]

We have performed (twice) *HeT-A* RNA FISH on *vasD1/D1; nosGAL4>GFP-VasWT* and *vasD1/D1; vasGAL4>GFP-VasWT* ovaries (Reviewer Figure 1A). As expected (and pointed out by the reviewer), *HeT-A* RNA signal was not detected in oocytes of *vasD1/D1; vasGAL4>GFP-VasWT* flies (Reviewer Figure 1A), although it was detected in the nurse cell nuclei as in wild-type (Figure 5B). Although *vasD1/D1; nosGAL4>GFP-VasWT* oocytes and nurse cell cytoplasm do contain *HeT-A* RNA, the RNA is detected in scant foci and does not resemble the *HeT-A* RNA pattern in *vas,mnk* double mutants, which display a massive accumulation of *HeT-A* RNA in both the nurse cells and the oocyte. As we do not think these data add to our understanding of the transgenerational effect of Vasa-dependent transposon deregulation, we have not included them in the manuscript. However, if the reviewers think the the data should be included, we could add them as a Supplementary Figure.

Reviewer Figure 1. Localization of *HeT-A* RNA and HeT-A/Gag protein.

(A) *In situ* detection of *HeT-A* mRNA by FISH on stage 9 egg-chambers of *vas^{D1/D1}; nos-Gal4>GFP-Vas^{WT}* and *vas^{D1/D1}; vas-Gal4>GFP-Vas^{WT}* flies. Arrows indicate sites of *HeT-A* mRNA transcription. Arrows indicate *HeT-A* RNA signal. Scale bars indicate 25 μ m and 5 μ m (5x magnification).

(B) Immunohistochemical detection of HeT-A/Gag protein in on early stage egg-chambers of wild-type (*w¹¹¹⁸*), *vas^{D1/D1}*, *vas^{D1/D1}; nos-Gal4>GFP-Vas^{WT}*, and *vas^{D1/D1}; vas-Gal4>GFP-Vas^{WT}* flies. Arrows indicate HeT-A/Gag signal. Scale bar indicate 20 μ m.

6. The authors should show the localization and expression pattern of HeT-A RNA and HeT-A/Gag protein in ovarioles of nosGAL4 and VasGAL4 ovarioles, especially between stages 2 and 6 to

show how early are these transposons invade the oocyte and transient loss of *vas* is sufficient for this inheritance. [MO: do or rebuttal]

We performed immunohistochemical detection of HeT-A/Gag protein in *vasD1/D1; nosGAL4>GFP-VasWT* and *vasD1/D1; vasGAL4>GFP-VasWT* ovaries (Reviewer Figure 1B). As expected HeT-A/Gag signal was not detected in *vasD1/D1; vasGAL4>GFP-VasWT* egg-chambers, and although we could detect HeT-A/Gag in *vasD1/D1; nosGAL4>GFP-VasWT* egg-chambers, the protein localized mainly in the nurse cell nuclei. As for the previous point, we do not think these data enhance our understanding of the transgenerational effect of Vasa-dependent transposon deregulation, therefore we have not included them in the manuscript. However, if the reviewers think the the data should be included, we could add them as a Supplementary Figure.

Minor points:

1. The authors should not say *nanos* promoter was inactive between stages 2 and 6 as they have not shown this. A better word would be "attenuated". [MO: edit]

We have changed “inactive” to “attenuated” (Line105).

Referee 2

The manuscript by Durdevic et al. reports two interesting observations: (1) Transient loss of *vasa* expression during early oogenesis (stages 2-6) leads to transposon up-regulation in the germline, which does not affect the progression of oogenesis but leads to increased DNA damage in egg chambers and resulting early embryos. (2) A *Chk2* mutation restores oogenesis of *vasa* mutant females without silencing transposons. These observations add to our understanding of the role of *Vasa* in transposon silencing and oogenesis. However, given the known role of *Vasa* in the piRNA pathway and the known role of the piRNA pathway in oogenesis (e.g. Klattenhoff et al. 2007), the findings reported in this manuscript are largely confirmatory. They bear limited biological novelty. In addition, the manuscript lacks molecular/mechanistic insights to key issues such as the nature of *Ch2* epistasis over *vasa*. For example, the author did not even attempt to examine whether the *Chk2* mutation changes the phosphorylation of *Vasa* as previously reported by Klattenhoff et al. (2007). Lastly, some results are weak and some interpretations are either biased or over-stated (see below). Therefore, the manuscript not suitably for publication in EMBO Report at this stage.

Specific comments:

(1) Fig 1A and p4, para 2, the author stated that "when driven by *nos-Gal4*, *GFP-VasWT* had no effect on transposon levels", and they attributed this to "transposon de-repression that lack of *Vasa* between stages 2 and 6 of oogenesis". It is difficult to rationalize why stages 2-6 are so crucial for transposon silencing. Did the authors make sure that the qPCR is completely linear to the RNA levels? A nonlinear qPCR result (which often is the case unless the reaction is well-calibrate) may preferentially reduce the difference between the levels of two RNA samples that are both well above the detection limit. This could be the case for the *vas* mutant ovaries and the *nos-Gal4>GFP-vasa* ovaries. In support of this possibility, the transgenic ovaries did display reduced transposon RNA levels (unlike the authors' claimed no difference), even though the reduction may not be statistically significant (?). [MO: address or rebuttal]

We have addressed the reviewer's concern about linearity of qPCR by analysing changes of Ct (Cycle threshold) values of three RNAs in a dilution series of samples of flies of the four genotypes analysed in Figure 1A. Our analysis shows that in all cases the qPCR was linear, as apparent from the R² values presented in Reviewer Figure 2. We further agree with the reviewer that transgenic ovaries display slightly reduced transposon levels compared to the mutant presented in Figure 1A, however as the reduction is not statistically significant we consider these transposons as not silenced.

Reviewer Figure 2. Regression plots showing linearity of RT-qPCR for *rp49* (A), *Burdock* (B) and *HeT-A* (C). RNA amounts are represented in a 10-fold dilution series (0.005, 0.05, 0.5, 5 and 50 ng).

(2) Fig 1A: It seems that Vasa is more effective in silencing *Het-A* (non-LTR) transposons than LTR transposons. The authors need to point out this and ideally also provide an Explanation. [MO: address]

A differential effect of Vasa on LTR – non-LTR transposon silencing is only observed if we compare *Burdock* and *HeT-A*, but not *blood* (another LTR transposon) and *HeT-A*, as shown in the Figure 1A. Therefore we think that it is not a general differential effect of Vasa on LTR vs. non-LTR transposons, but rather an effect on a specific transposon. This might be due to the steady-state levels of expression and copy numbers in the genome of the respective transposons (*Burdock* is present in 6411 copies, whereas *blood* in only 22 copies (Kaminker et al., *The transposable elements of the Drosophila melanogaster euchromatin: a genomics perspective. Genome Biol. 3(12): RESEARCH0084. 2002*)).

(3) P5, para 1: "The fact that in spite of transposon up-regulation oogenesis and egg-laying rates were fully restored in *vasD1/D1* flies (Figure 1A, indicated by + and - and Supplementary Figure S1D-F) indicates that oogenetic processes are not affected below a certain threshold of transposon activity." These statements are inaccurate and somewhat misleading. First of all, the egg laying rate is not fully, but only largely, restored, with *nos-Gal4>GFP-vasa* females showing less restoration than *vas-Gal4>GFP-vasa* females. This is consistent with the notion that transposon activation affects oogenesis but does not completely block it, unless the level of activation is extremely high to cause ovarian Dysgenesis. [MO: address by text edit or rebuttal]

We changed “fully” to “largely” in the revised manuscript and added a clause inspired by the reviewer’s comment, such that the text now reads: “The fact that in spite of transposon up-regulation oogenesis and egg-laying rates were largely restored in *vasD1/D1* flies (Figure 1A, indicated by + and – and Supplementary Figure S1D-F) is consistent with the notion that transposon activation affects but does not completely block oogenesis unless the level of activation is so high as to cause its arrest.” (Line 122-126).

We hope that our inclusion of this edit suggested by the reviewer is acceptable.

BTW, "oogenetic" should be "oogenic". [MO: fix]

We have excluded the term “oogenetic” (Lines 122-126) in the revised sentence cited above.

(4) Do *vasD1/D1* females lay eggs at all? This needs to be reported and included in Fig S1E, even if the mutant females do not lay eggs. If eggs are laid by the mutant, Fig 2E need to include a 0-1 hour *vasa* mutant embryo control to compare the levels of *HetA/Gag* protein in these embryos with those in *nos-Gal4>GFP-vasa* and *vas-Gal4>GFP-vasa* embryos. Otherwise, it is difficult to make a proper conclusion from the western blot. In addition, the protein loading of different lanes needs to be more uniform to allow more meaningful comparison. Likewise, Fig3A and other relevant figures also need to include a control of 0-1 hour *vasa* mutant embryos. [MO: fix and add better western]

The *vasa* null (*vasD1/D1*) females undergo an early oogenesis arrest and do not lay eggs, therefore it is not possible to perform experiments shown in Figure 1C and D, Supplementary Figures S1E, S3C and S4A. To make this point clear, in the revised manuscript we have added the sentence (Lines 134-136): “Embryos from *vas^{D1/D1}* mutant flies could not be included in these and all the other experiments on embryos, as *vas^{D1/D1}* females arrest oogenesis early and do not lay eggs.” We have also added new western blots in Figure 2E and Supplementary Figure S4C, as requested by the reviewer.

(5) P7, line 4, "Supplementary Figure S3B panel b" should be "Supplementary Figure S3A panel b". [MO: fix]

We apologize for the confusion and have corrected this in the revised manuscript (Lines 181-186).

(6) P9, Discussion: "we have shown that, independent of Chk2 signaling, de-repressed transposons are responsible for nuclear damage and embryonic lethality." This is an over-statement. The authors established a solid correlation, but have not definitively shown, that the de-repressed transposons are responsible for nuclear damage and embryonic lethality. [MO: edit or rebuttal]

We appreciate reviewer’s remark and in the revised manuscript have modified the sentence (Lines 252-257) such that it now reads: “By tracing transposon protein and RNA levels and localization from the mother to the early embryos we suggest that, independent of Chk2 signaling, de-repressed transposons are responsible for nuclear damage and embryonic lethality.”

Referee 3

The authors study the effects of *vasa* mutations on oogenesis as well as embryogenesis of *Drosophila*. The authors find that depletion of the fly Chk2 ortholog *mnk* restores oogenesis in *vas* mutants, but does not suppress defects in transposon silencing or DSB-induced nuclear damage and embryonic lethality. This paper suggests that the *Vasa*-dependent protection against transposons in the nurse cells is essential to prevent DNA damage-induced arrest of embryonic development. The novelty of the present study however has been dampened by previous studies by others demonstrating that depletion of the *mnk* restores oogenesis in mutations in piRNA pathway components (Chen et al., 2007; Klattenhoff et al., 2007; Pane et al., 2007; and others). In addition, Haifan Lin and colleagues made double mutants of *mnk* and *ago3*, a component of the piRNA pathway, to examine the role of DNA damage signaling in *Ago3*-depleted embryo defects and found that while egg laying was rescued in the double mutants, egg hatching remained defective (see Figure 6 in the paper of Mani et al., 2014). Overall, with a few relatively small exceptions, this paper mostly repeats data already published by other groups.

Other comments:

1. Only Figure 1A and Figure 3B show the results with *vas[D1/D1]*. The authors should show the results with *vas[D1/D1]* in all other experiments (in particular, Fig. S1E, Fig. 1C & D, Fig. S3C, and Fig. S4A) as controls. [MO: add if possible]

As indicated in our response to Reviewer 2’s point 4, the *vasa* null (*vasD1/D1*) females undergo an early oogenesis arrest and do not lay eggs, therefore it is not possible to perform experiments shown in Figure 1C and D, Supplementary Figures S1E, S3C and S4A. We have included a sentence clarifying this point in the revised manuscript (Lines 134-136).

2. Figure 2A: It is unclear how the authors define "nuclear damage." [MO: address]

We qualify embryos as having nuclear damage when they contain nuclei with a misshapen, aberrant morphology, compared to the nuclei of wild-type embryos. In the revised manuscript we have clarified this point by rewriting the sentence such that it now reads (Lines 154-157): “Quantification of embryos containing nuclei of aberrant nuclear morphology (Fig 2A lower panel) compared to the nuclei of wild-type embryos (Fig 2A upper panel), revealed a high proportion of such nuclear defects among *vas^{D1/D1}*; *nos-Gal4*>GFP-*Vas^{WT}* embryos (Fig 2A).”

Thank you for submitting your revised manuscript entitled "Transposon silencing in the *Drosophila* female germline ensures genome stability in progeny embryos" along with your response to the referee concerns. We have now looked at both documents and we would be happy to publish your paper in Life Science Alliance pending final revisions necessary to meet our formatting guidelines.

Before we can go on to officially accept the study we would therefore ask you to address the following minor points in a final revised version:

- > Please include a brief call-out to the table of primer-sequences in the main text or provide a description of the content in the file title/as a legend
- > Table S7 is called out in the ms but file not provided with the submission
- > Please include call-outs for Tables S3 and S6
- > Please specify the origin of magnifications in Fig1B
